# Integrating the OHIF Viewer into XNAT: Achievements, Challenges and Prospects for Quantitative Imaging Studies

Simon J. Doran [1,2,*], Mohammad Al Sa'd [2,3], James A. Petts [4], James Darcy [1,2], Kate Alpert [5], Woonchan Cho [6], Lorena Escudero Sanchez [2,7,8], Sachidanand Alle [9], Ahmed El Harouni [9], Brad Genereaux [9], Erik Ziegler [10,11], Gordon J. Harris [10,12,13], Eric O. Aboagye [2,3], Evis Sala [2,7,8], Dow-Mu Koh [2,14] and Dan Marcus [5,6]

[1] Division of Radiotherapy and Imaging, Institute of Cancer Research, 15 Cotswold Rd, London SM2 5NG, UK; james.darcy@icr.ac.uk

[2] CRUK National Cancer Imaging Translational Accelerator, UK; m.alsad@imperial.ac.uk (M.A.S.); les44@cam.ac.uk (L.E.S.); eric.aboagye@imperial.ac.uk (E.O.A.); es220@cam.ac.uk (E.S.); mu.koh@icr.ac.uk (D.-M.K.)

[3] Cancer Imaging Centre, Department of Surgery & Cancer, Imperial College, London SW7 2AZ, UK

[4] Ovela Solutions Ltd., 20-22 Wenlock Road, London N1 7GU, UK; jamesapetts@ovelasolutions.com

[5] Flywheel LLC, 1015 Glenwood Ave, Suite 300, Minneapolis, MN 55405, USA; katealpert@flywheel.io (K.A.); dmarcus@wustl.edu (D.M.)

[6] Neuroimaging Informatics Analysis Center, Washington University School of Medicine, 660 S Euclid Ave, St. Louis, MO 63110, USA; wcho24@wustl.edu

[7] Department of Radiology, University of Cambridge, Hills Rd, Cambridge CB2 0QQ, UK

[8] Cancer Research UK Cambridge Centre, University of Cambridge Li Ka Shing Centre, Robinson Way, Cambridge CB2 0RE, UK

[9] NVIDIA, 2788 San Tomas Expressway, Santa Clara, CA 95051, USA; salle@nvidia.com (S.A.); aharouni@nvidia.com (A.E.H.); bgenereaux@nvidia.com (B.G.)

[10] Open Health Imaging Foundation, Massachusetts General Hospital, 55 Fruit St., Boston, MA 02114, USA; erik.ziegler@radicalimaging.com (E.Z.); harris@helix.mgh.harvard.edu (G.J.H.)

[11] Radical Imaging LLC, 188 Annie Moore Rd, Bolton, MA 01740-1140, USA

[12] Department of Radiology, Massachusetts General Hospital, 55 Fruit St., Boston, MA 02114, USA

[13] Harvard Medical School, 25 Shattuck St., Boston, MA 02115, USA

[14] Department of Radiology, Royal Marsden Hospital, Downs Rd, Sutton SM2 5PT, UK

* Correspondence: simon.doran@icr.ac.uk; Tel.: +44-208-661-3718

**Abstract:** *Purpose*: XNAT is an informatics software platform to support imaging research, particularly in the context of large, multicentre studies of the type that are essential to validate quantitative imaging biomarkers. XNAT provides import, archiving, processing and secure distribution facilities for image and related study data. Until recently, however, modern data visualisation and annotation tools were lacking on the XNAT platform. We describe the background to, and implementation of, an integration of the Open Health Imaging Foundation (OHIF) Viewer into the XNAT environment. We explain the challenges overcome and discuss future prospects for quantitative imaging studies. *Materials and methods*: The OHIF Viewer adopts an approach based on the DICOM web protocol. To allow operation in an XNAT environment, a data-routing methodology was developed to overcome the mismatch between the DICOM and XNAT information models and a custom viewer panel created to allow navigation within the viewer between different XNAT projects, subjects and imaging sessions. Modifications to the development environment were made to allow developers to test new code more easily against a live XNAT instance. Major new developments focused on the creation and storage of regions-of-interest (ROIs) and included: ROI creation and editing tools for both contour- and mask-based regions; a "smart CT" paintbrush tool; the integration of NVIDIA's Artificial Intelligence Assisted Annotation (AIAA); the ability to view surface meshes, fractional segmentation maps and image overlays; and a rapid image reader tool aimed at radiologists. We have incorporated the OHIF microscopy extension and, in parallel, introduced support for microscopy session types within XNAT for the first time. *Results:* Integration of the OHIF Viewer within XNAT has been highly successful and numerous additional and enhanced tools have been created in a programme started in 2017 that is still ongoing. The software has been downloaded more than 3700 times during the course of the development work reported here, demonstrating the impact of the work. *Conclusions*: The OHIF

open-source, zero-footprint web viewer has been incorporated into the XNAT platform and is now used at many institutions worldwide. Further innovations are envisaged in the near future.

**Keywords:** XNAT; OHIF; web viewer; image visualisation; regions-of-interest; rapid reader

## 1. Introduction

Clinical imaging scanners produce data that are stored in a metadata-rich and highly standardised format (Digital Imaging and Communications in Medicine, DICOM). Within the normal clinical workflow, these images are visualised using commercial Picture Archiving and Communications Systems (PACS). However, research studies involving imaging entail additional data handling and visualisation requirements that are not well catered for by PACS.

This article describes novel work to integrate the Open Health Imaging Foundation (OHIF) Viewer [1,2] into the research platform XNAT [3]. We set out the context of the work and explain the design challenges, critically assessing the role of the contrasting data models used by DICOM (as inherited by PACS and OHIF) and XNAT. We outline our methodology and achievements to date, demonstrating significant progress over the work reported in previous publications. Finally, we discuss the advantages and limitations of the viewer and the prospects for future development, from both a technical and radiologist perspective. Our work focuses on image annotation since a key requirement for progress in the burgeoning field of artificial intelligence (AI)-assisted healthcare is an improvement in the quantity and quality of annotated data available for model training and validation.

### 1.1. Conceptual Differences between XNAT and PACS

XNAT [3] is a cross-platform, open-source tool from Washington University, St Louis, designed specifically to support imaging research. Its core function is to manage the import, archiving, processing and secure distribution of image and related study data. XNAT fulfils the following research needs, for which facilities are usually absent on commercial PACS for governance reasons and because PACS addresses a different use case:

- Research data should be normally curated in "projects" that reflect identifiable academic activities (e.g., clinical trial, PhD project, blinded image review, online analysis "challenge") each of which may have an individual Data Management Plan.
- Researchers from many different organisations (e.g., hospital, academia, industry) may need to access the platform.
- Unlike PACS, where any clinical user might need to access images for any patient, user permissions are customised per project according to ethical protocols, data transfer agreements, collaborations and time-limited embargoes.
- Academic principal/chief investigators may demand a high degree of autonomy, with the ability to curate, structure and manage their own information assets.
- Research data require anonymisation prior to introduction into the academic workflow and this may need to be tailored to individual studies.
- The repository platform should make data findable, accessible, interoperable and reusable (FAIR) [4]. This is typically achieved by equipping platforms with Representational State Transfer (REST) Application Programming Interfaces (APIs), thus enabling integration with a diverse range of end-user tools [5].
- Projects may combine DICOM with clinical, digital pathology, multi-omic and other non-DICOM data. Each project may also be associated with its own bespoke analysis software and data formats.
- Arbitrary processing outputs are frequently created by external data analysis tools and need to be stored back on the repository platform with appropriate provenance.

- Data enrichment via expert annotation should be exportable and should use standardised, portable formats, rather than be "locked into" a given vendor's image display platform.
- Most importantly, for the remainder of this article, the processing and data visualisation methods used are often the subject of the research itself. Hence, the image viewer configuration needs to be agile, with the potential for incorporation of novel software that is, by definition, experimental and has not undergone regulatory approval for clinical use.

### 1.2. Image Viewing in XNAT, the OHIF Viewer and Other Web-Based Solutions

XNAT has a significant track record, being chosen, for example, as the archiving platform for the Human Connectome Project [6] but, until around 2015, it was distributed with only limited built-in capabilities for data visualisation (thumbnail images). With XNAT 1.6.5, this changed via the introduction of XImgView, an HTML5 web-based image viewer, built on XTK, a WebGL toolkit for scientific visualisation [7]. That technology, however, proved difficult to develop further and, within a few years, the need for a more "PACS-like" viewing solution became apparent.

The Open Health Imaging Foundation (OHIF) [2] is a consortium of academic and commercial partners with a shared vision to create open-source components, allowing rich medical imaging applications to be built with far less effort than would be needed to create a fully featured product from scratch. The eponymous "OHIF Viewer" [1] is the most high-profile output: it collects together a number of underlying libraries, developed by the consortium, into a zero-footprint, browser-based, PACS-like viewer that can be customised and "white-labelled".

Other web viewers have also been developed. Wadali et al. [8] followed the PRISMA guidelines [9] to survey the overall landscape in 2020. They found more than 200 DICOM viewer projects underway, but only six zero-footprint viewers that met their most important criteria, with the GitHub project DWV [10] matching their use case best. One of the closest comparators for our quantitative imaging application is ePAD [11], which combines an image viewer and annotation capabilities with the DCM4CHEE [12] image archive. The platform implements templates from the AIM (Annotation and Imaging Markup) Template Builder [13] and was initially an attractive candidate for our quantitative studies. However, the version available at the time of our evaluation in 2016 was insufficiently user-friendly for busy radiologists. Contemporaneous with our work and very recently documented in the literature [14], Studierfenster is a non-commercial open science client-server framework for (bio-)medical image analysis, with a focus on neuroimaging and augmented reality. Two additional web applications, Biomedisa [15] and Gradio Hub [16], whilst not being general-purpose medical image viewers like OHIF, overlap somewhat with the segmentation and AI aspects, respectively, of the work presented below.

### 1.3. Quantitative Imaging Motivation for Development of the OHIF Viewer within XNAT

The unique and novel feature of the programme of work described here is the combination of a zero-footprint web-based viewer (OHIF) with an already state-of-the-art research data management and archiving tool (XNAT), together with their application in the field of quantitative imaging clinical trials.

The Institute of Cancer Research (ICR) and eight other UK universities (Cambridge, Glasgow, Imperial College London, Kings College London, Manchester, Newcastle, Oxford and University College London) work together to deliver clinical imaging studies and trials within the context of the National Cancer Imaging Translational Accelerator (NCITA), funded by Cancer Research UK, the UK's largest cancer charity [17]. A primary focus of NCITA is to accelerate the adoption of quantitative imaging biomarkers into clinical practice, a goal that is shared with the US National Cancer Institute (NCI) Quantitative Imaging Network (QIN). One aspiration is to develop "locked down pipelines" for the generation of imaging biomarkers that can be qualified in imaging studies [18]. XNAT's

extensibility helps to achieve this through its Container Service which allows us to incorporate external tools (for example, novel image reconstruction algorithms, or analysis methods like QIN's pyRadiomics [19]) through its Container Service. Tools can be invoked interactively from the viewer or XNAT report pages, invoked programmatically via REST API calls, or "triggered" on "events" such as data archival. This last possibility allows results to be computed in the background without intervention, in advance of a radiologist reporting session.

### 1.4. Image Annotation

Ziegler et al. [1] commented that " . . . the most commonly requested feature for OHIF has been robust configurable segmentation tools to allow clinicians and researchers to review and correct machine-generated label maps". The definition of regions-of-interest is also a prerequisite for the majority of quantitative imaging pipelines. The work described below comprehensively achieves this and ensures that the OHIF viewer is well suited for the type of studies undertaken by NCITA and QIN, particularly for the validation and translation of novel imaging tools and biomarkers from the research domain into clinical trials and patient care.

## 2. Materials and Methods

### 2.1. Integration Timeline

An overview schematic of the project to date is depicted in Figure 1. Initial discussions on the potential use of OHIF took place in 2016–2017. One important catalyst was the release in September 2016 of XNAT 1.7: the new plugin infrastructure made it much easier for members of the XNAT community outside the core team at Washington University to develop and distribute new functionality without needing to modify the main XNAT webapp. Releases 1 and 2 of the viewer integration, developed at ICR, both consisted of two plugins, one containing primarily the "front-end" viewer code and a second implementing server-side schema changes and REST API endpoints to support the newly introduced ROI model. From Release 3.0 onwards, these have been combined to allow the viewer to be installed as a single plugin. Separation of the viewer from the main XNAT webapp allows for independent release cycles and makes customisation easier for sites with relevant expertise. By the time of the 1.0 release of the ICR-XNAT-OHIF viewer, key routing and data-access issues had already been resolved, allowing a simple click-through from XNAT's imaging session page, with the complete session pre-loaded in the standard OHIF sidebar.

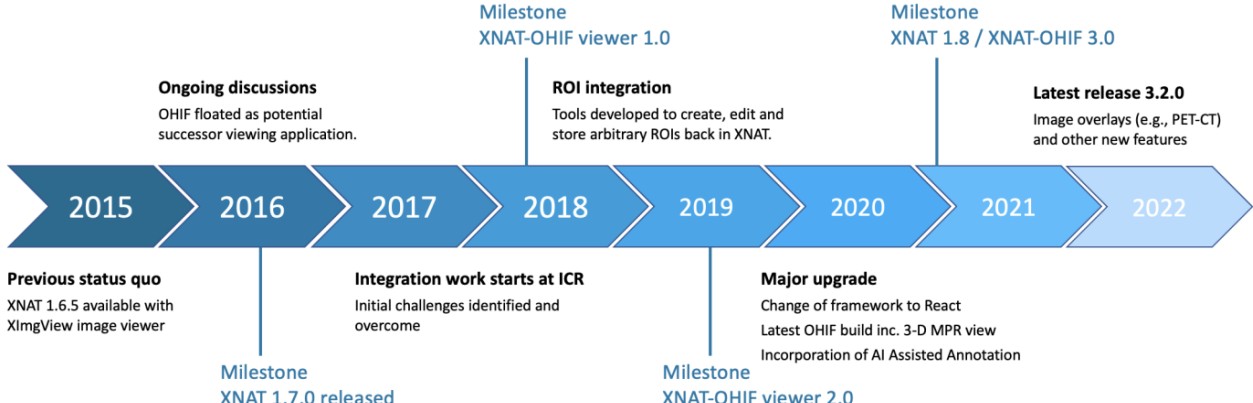

**Figure 1.** ICR-XNAT-OHIF viewer development project timeline.

An important limitation of the OHIF Viewer prior to our work was that, although annotation options were available (looking broadly similar to those available on a standard PACS), most were essentially "on-screen decoration" overlaid on the images and could be captured only as screenshots, not in a machine-readable format for analysis. A key piece of work was, therefore, to create client-side facilities to define and edit arbitrary

3D regions-of-interest (ROIs) and, on the server side (by creating new XNAT REST API calls), to archive these in XNAT's storage back-end as appropriate XNAT data types. This work, together with numerous usability improvements, culminated in the ICR-XNAT-OHIF Viewer 2.0 release in August 2018.

During this period, the OHIF consortium started a move from the Meteor web framework [20] to the more modern React [21]. Release 3.0 of our integration [22], launched simultaneously with XNAT 1.8, represented a significant overhaul of the ICR-XNAT-OHIF codebase and brought it in-line with the latest OHIF architecture, incorporating new features such as 3D MPR view and dramatically increasing the speed of image segmentation with support for Artificial Intelligence Assisted Annotation (AIAA).

### 2.2. Initial Challenges

An important issue to address at the outset was how to "feed" the OHIF viewer with the correct data from XNAT's image archive. Challenges arise from a significant divergence in the data model between DICOM and XNAT.

DICOM's information model [23] has a simple hierarchical structure of the form Patient → Study → Series → SOPInstance → Frame. The "Series" level comprises entities serving many different real-world purposes (images, annotations, registrations, waveforms, radiotherapy objects, presentation states, etc.), so that a DICOM "series" can, perhaps, best be regarded as a "wrapper". A single DICOM study may contain a series of multiple types. To retrieve a data object from PACS, one typically needs only its so-called SOPInstanceUID, although combining this with study and series UIDs may allow for more efficient searching.

XNAT has a more complex inheritance model [3]. In broad terms, XNAT subjects correspond to DICOM patients (but with the change of terminology explicitly recognising the fact that clinical patients represent only one type of research subject, others being volunteers, phantoms or preclinical subjects). XNAT sessions are the equivalent of DICOM studies and XNAT scans are like DICOM series. However, each different imaging modality has a different XNAT session type. This allows different session metadata to be stored in XNAT's PostgreSQL database, permitting rapid and highly customisable searching for imaging sessions matching particular criteria. XNAT's database schema relates to concepts of data origin and data purpose in a very different way from that of DICOM: thus, for example, image segmentations are simply additional series in the DICOM information model, whereas they are ROI collections in XNAT's schema, inheriting from image assessors in a way that emphasises the fundamentally different purpose of a post-acquisition image interpretation step compared with raw image data acquisition.

XNAT's concept of a "project"—the top level of the storage hierarchy in the XNAT repository structure—has no direct equivalent in DICOM. A given patient or DICOM study can be imported into more than one project and this means that a given DICOM SOPInstance may be found in more than one place within the repository. This presents complications to the implementation of the DICOMweb protocol [24] (the OHIF Viewer's preferred mechanism for data retrieval) as no one-to-one mapping between SOPInstanceUID and image Uniform Resource Identifier (URI) exists in the general case. Furthermore, in a situation rarely encountered on PACS, a security model governs access to each XNAT data object, with granular access to the different data types described above.

Access to the viewer is enabled at both subject and session levels, allowing images of the patient at different time points to be compared. Integration of the viewer with XNAT's hierarchy is also reinforced by a new "XNAT navigation" panel.

Although XNAT's DICOMweb support is now in late beta development, it was not in place at the start of this work. Fortunately, the OHIF Viewer provides a second mechanism for specifying image data locations: a list of the DICOM Part 10 (P10) files comprising each image session, together with key metadata required by the viewer is stored in JavaScript Object Notation (JSON) format using the XNAT Configuration Service. Such listings are relatively time-consuming to construct for large imaging sessions and initial versions of the integration suffered from significant delays in opening images on the first view. Plugin

releases after 2.1 use the XNAT event service to trigger automatic background calculation of this "session JSON" when images are imported into XNAT.

This methodology has been highly successful and, from an end-user perspective, integration in the 2.x and 3.x series of releases of the ICR plugin has been seamless. However, from an architectural point of view, there are fundamental differences from "core-OHIF" (i.e., the un-customised open-source OHIF product), both in the user interface (UI) and "under the hood".

- Accessibility of images is governed by the XNAT security model.
- Non-image DICOM series (e.g., ROIs) are removed from the study thumbnail list and handled by separate elements of the user interface (the contour and mask panels).
- Storage of ROIs uses XNAT's REST API, creating new session resources and XNAT assessors, rather than simply adding contours and masks as new DICOM series in the study.
- Several other tools (e.g., the XNAT navigation sidebar and integration of the AIAA server) make calls to the XNAT REST API.

The ICR-XNAT-OHIF integration thus involves more than simply making a backend that will "serve" DICOM data to the standard OHIF viewer. Consequently, it has not been possible to create the integration as an OHIF "extension" [25]. The code base must currently be "forked" and this increases the developer effort required to maintain version parity with core-OHIF. We expect the problem to be alleviated when the next major version of core-OHIF (known as "OHIF-v3", but which should be carefully distinguished from Release 3 of the ICR-XNAT-OHIF integration) fully implements the new concept of "modes" [26].

*2.3. Development Environment*

The standard OHIF viewer can be configured to connect, even when running within an Integrated Development Environment (IDE), to any DICOMweb-compliant data source, for example, an Orthanc imaging archive [27]. However, as discussed earlier, XNAT did not originally support DICOMweb and running a debugger session for the viewer code against a live XNAT instance was thus not possible. Early front-end development required an extensive rebuild cycle that involved shutting down XNAT's Apache Tomcat webserver and redeploying a new plugin JAR file and restarting XNAT. The whole process takes approximately 15 min each time code is changed, can be subject to errors and is thus an inefficient strategy. To mitigate these issues, we have implemented new functionality that connects a JavaScript IDE directly to a live XNAT instance.

*2.4. Regions-of-Interest*

As noted above, machine-readable annotation has been a major focus of our work to date. The rise of both machine learning (ML) in general and radiomics [28,29] in particular have created a demand to store and exchange ROIs. These are traditionally represented either as stacks of contours, widely used in radiotherapy, or masks, which are the commonest format for ML and might be binary, or multi-valued label-maps representing different tissue classes, or so-called "fractional segmentations", typically representing probability maps. Triangular meshes describing surfaces are also found in more specialised medical applications, often in computational modelling. DICOM provides Information Object Definitions (IODs) for all three types, but initially, we focused on the RT Structure Set IOD for contours drawn on 3D data and the Segmentation IOD for masks. We faced the issue—made particularly acute by the need to annotate digital X-ray images during the COVID-19 pandemic—that there is currently no widely adopted DICOM IOD for describing 2D ROIs (although a technical whitepaper [30] describes a mechanism for representing these within a DICOM Structured Report). We adopted the Annotation and Image Markup standard [13] as a pragmatic 2D solution for both digital x-rays (DICOM DX IOD) and mammograms (MG IOD).

Practical implementation of these capabilities involved the creation of two new React components (a contour panel and mask panel) together with a number of new tools to

allow users to define and manipulate ROIs. In response to user feedback, we developed specific tools for interpolation of contours between slices, and a "smart" CT paintbrush (filling only voxels satisfying defined criteria, based on configurable ranges of Hounsfield Units for CT, with auto-filling of small holes). Since Release 3, "fractional" DICOM segmentations have also been supported allowing end-users to represent probabilistic masks, as increasingly encountered in ML applications. Further customisations by Radiologics Inc. (now Flywheel), the manufacturer of the commercial version of XNAT, permit the representation of surface meshes in the form of the Standard Triangle Language (sometimes known as Standard Tessellation Language) STL files. (This material is not open-sourced at the time of writing).

### 2.5. AI-Assisted Annotation

NVIDIA's AI-Assisted Annotation is a software development kit (SDK) that provides client APIs, compatible with a range of viewers, to annotate structures in medical images. In-house NVIDIA experiments suggest the potential to decrease annotation times tenfold, whilst published data [31] have already demonstrated a 6.7-fold saving.

AIAA forms part of the Clara Application Framework [32], recently updated to incorporate the Medical Open Network for AI (MONAI) PyTorch-based open-source framework for deep learning in healthcare imaging [33]. AIAA comes with three types of organ-specific pre-trained models:

1. Fully automated "Segmentation Models" return multi-label segmentations without any user input.
2. Semi-automated "Annotation Models" use a minimum of six clicks from the user to define the bounding box of a structure and return the segmentation.
3. "Deep Grow Models" are interactive, taking a point of reference and using successive "foreground" and/or "background" clicks to refine the model's inference, by including or excluding regions.

With the Clara Train SDK, data scientists can not only fine-tune segmentation for these existing models using their own datasets but also create entirely new models. Clara represents just one example of integration. In future, we expect to see the introduction of other algorithms, tools and workflows into the viewer, and several of the tools developed through QIN [34]—particularly those that can be containerised or whose algorithms can be otherwise abstracted—will be good candidates.

### 2.6. Rapid Reader

Rapid Reader is an XNAT webapp that facilitates efficient navigation through DICOM sessions, pre-specified as worklists, and provides electronic case report forms (eCRF) to evaluate them. The concept of a worklist is familiar to radiologists, via their use in clinical PACS, and each XNAT worklist has properties that determine the visualisation and actions available to the user. Radiologists perform their "reads" on DICOM sessions serially and complete the eCRF in a new side panel of a customised viewer. Collected data are stored as an XNAT image assessor for further analysis.

The independent UI of the Rapid reader makes it more straightforward for a user to navigate assigned sessions without prior knowledge about XNAT or its data model (project, subject and experiment). The unity of visual identity between the worklist and viewer creates a more "PACS-like" experience than the standard route through the main XNAT webapp, and we anticipate that this will prove more acceptable to radiologists. The Rapid Reader plugin consists of both front-end code (navigation flow, custom tool specification and evaluation form panel) and back-end code (REST APIs and business logic to manage the worklists).

### 2.7. Other New Features

In our most recent release currently under development, we have further augmented the facilities of the standard OHIF viewer with the capability to overlay multiple image

modalities. The most recent versions of XNAT have added the DICOM GM and SM IODs to the list of modalities "understood" and, thanks to XNAT's new DICOMweb capabilities, we have been able to include the OHIF microscopy extension described in [1], to an alpha version of a future release.

## 3. Results

The integration of OHIF and XNAT open-source platforms to create the ICR-XNAT-OHIF viewer has attracted substantial interest (3729 downloads of the various plugin versions at the time of writing), which suggests a rapid take-up by XNAT sites. In this section, we present here several example use cases of the integrated framework.

Figure 2 illustrates the use of the XNAT viewer to visualise clinically segmented radiotherapy structures via support for DICOM's RT-STRUCT IOD. A single RT-STRUCT may contain many 3D regions of interest, corresponding to large numbers of contours visualised on multiple image slices. The viewer is not intended to function as a complete radiotherapy workstation, but simple facilities are implemented such as the selective viewing of different organs.

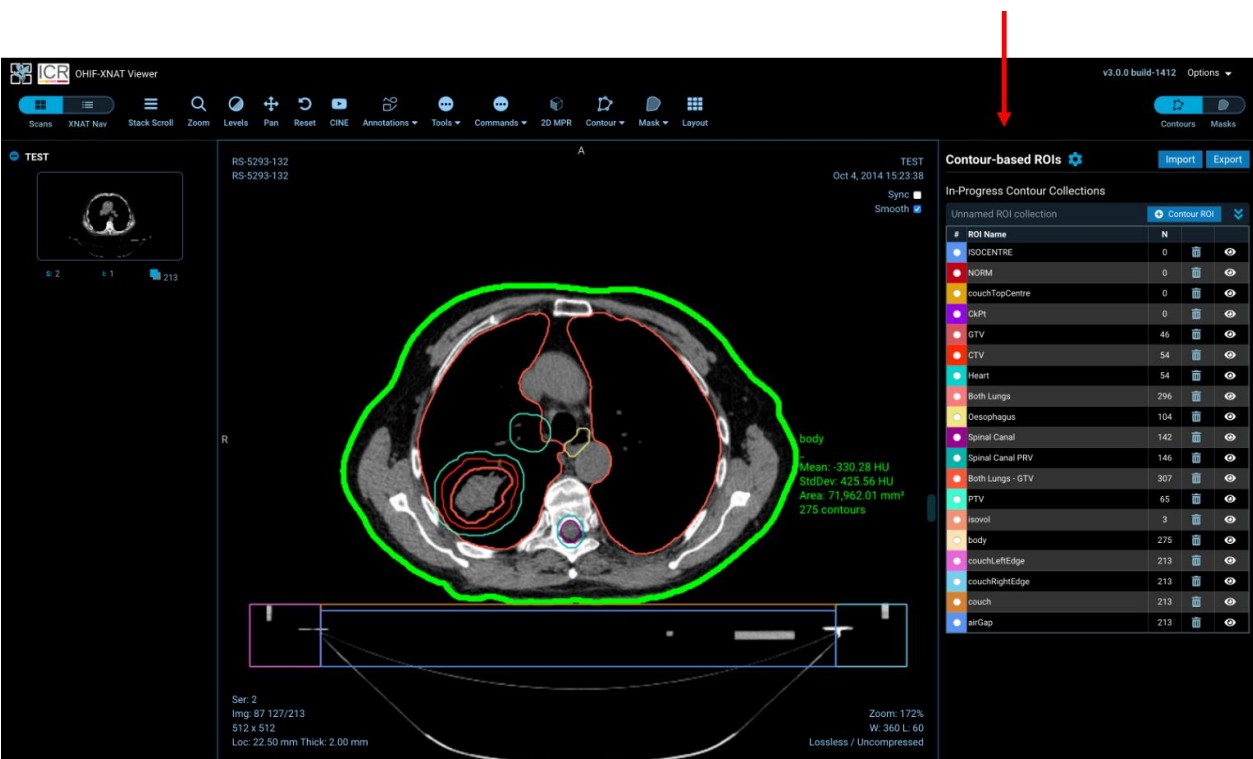

**Figure 2.** Visualisation of an RT Structure Set within the ICR-XNAT-OHIF viewer, also demonstrating the contour sidebar component developed as part of this project.

Figure 3 shows the interface for the NVIDIA AIAA tool. In the XNAT integration, the viewer is agnostic as to the origin of segmentations: AIAA results can be refined using the previously described manual editing tools and saved within XNAT as DICOM segmentation objects using the Mask ROI sidebar (a custom component created as part of this work).

Figure 4 demonstrates the multiplanar reformatting capabilities of the web viewer, together with the facility to display the DICOM "fractional" segmentation data type, together with other XNAT-specific integrations, such as the project navigator sidebar.

Figure 5 illustrates the capability of the viewer to display surface meshes.

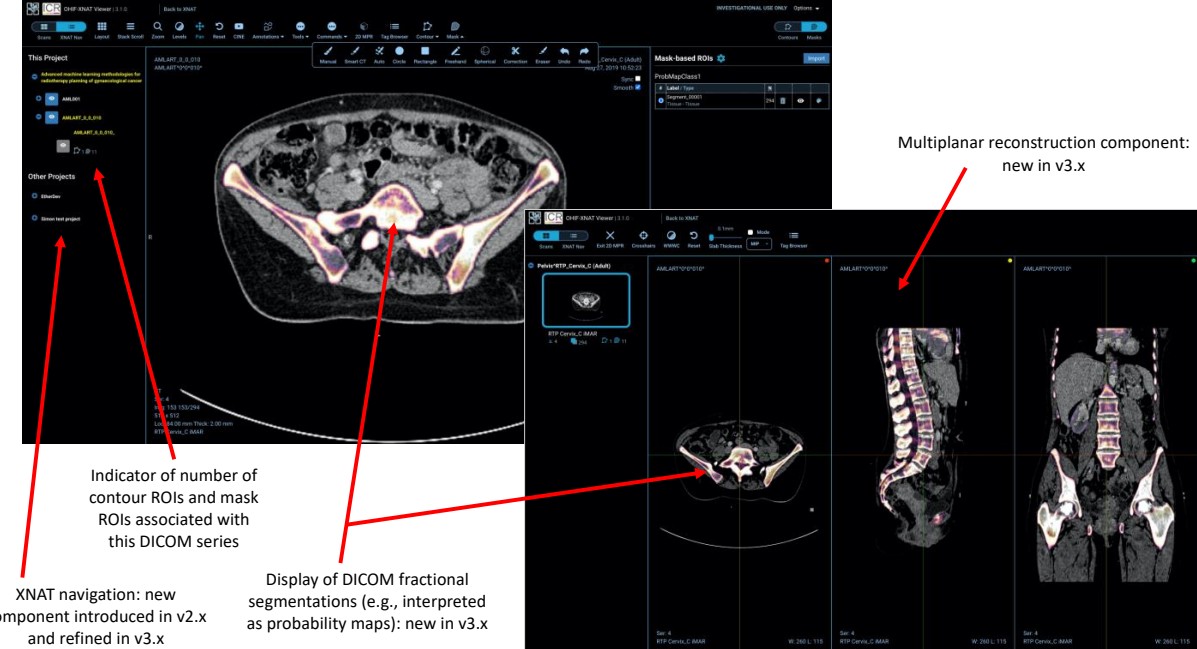

**Figure 3.** Our integration of the NVIDIA AIAA tool for automatic and semiautomatic segmentation based on machine learning models. A key advantage of the new tool is that the AI-assisted segmentations are processed and stored in exactly the same way as manual segmentations and so any shortcomings in the AI-based results, such as those seen in the left lung here can easily be refined manually and used to retrain the model.

**Figure 4.** Display of DICOM fractional segmentation objects both in standard 2D mode and multiplanar reformatting (MPR) mode, also demonstrating the integration of a new XNAT project navigation sidebar and that 3D mask ROIs are rendered correctly in all three planes.

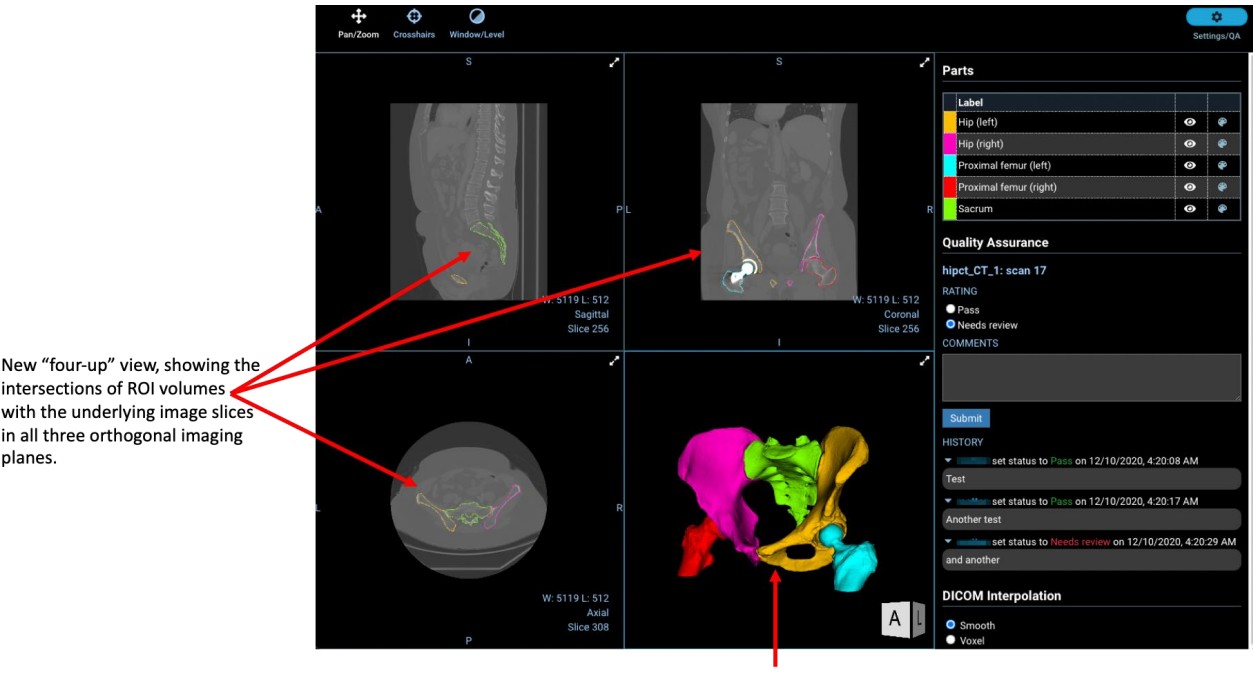

New "four-up" view, showing the intersections of ROI volumes with the underlying image slices in all three orthogonal imaging planes.

Ability to render STL files

**Figure 5.** Custom "four-up" view created by Radiologics Inc., demonstrating the visualisation of surface mesh files alongside contour-based ROIs. Currently available only commercially via Flywheel.

Figure 6 shows the new "composition" feature permitting image overlays to be used for the first time in a fully featured OHIF viewer. This is currently an XNAT-specific NCITA development based on the OHIF-v2 React but will be aligned with new core-OHIF overlay tools when they are fully mature.

New "composition" tool, enabling image overlays (e.g., PET-CT): new in forthcoming ICR-XNAT-OHIF release 3.2.0

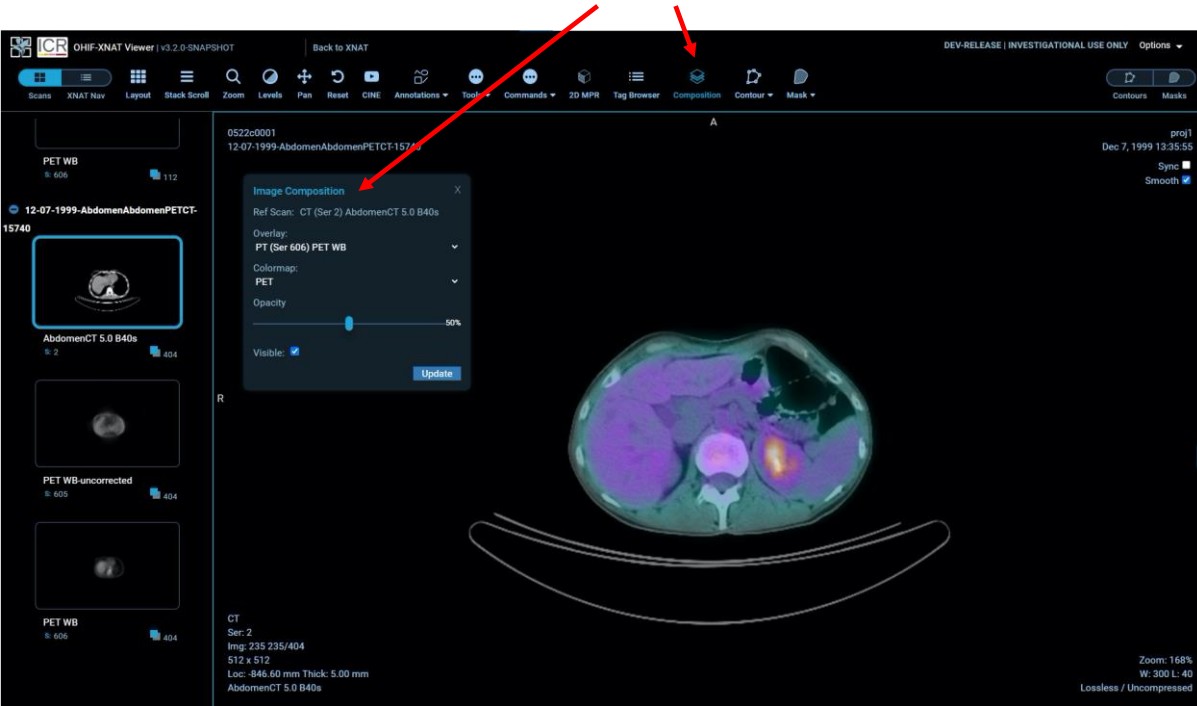

**Figure 6.** The new image composition tool, allowing overlay of DICOM series within a session to display, for example, PET-CT images.

Figure 7 illustrates the new Rapid Reader. The upper panel shows the worklist view and the lower panel the annotation view with eCRF.

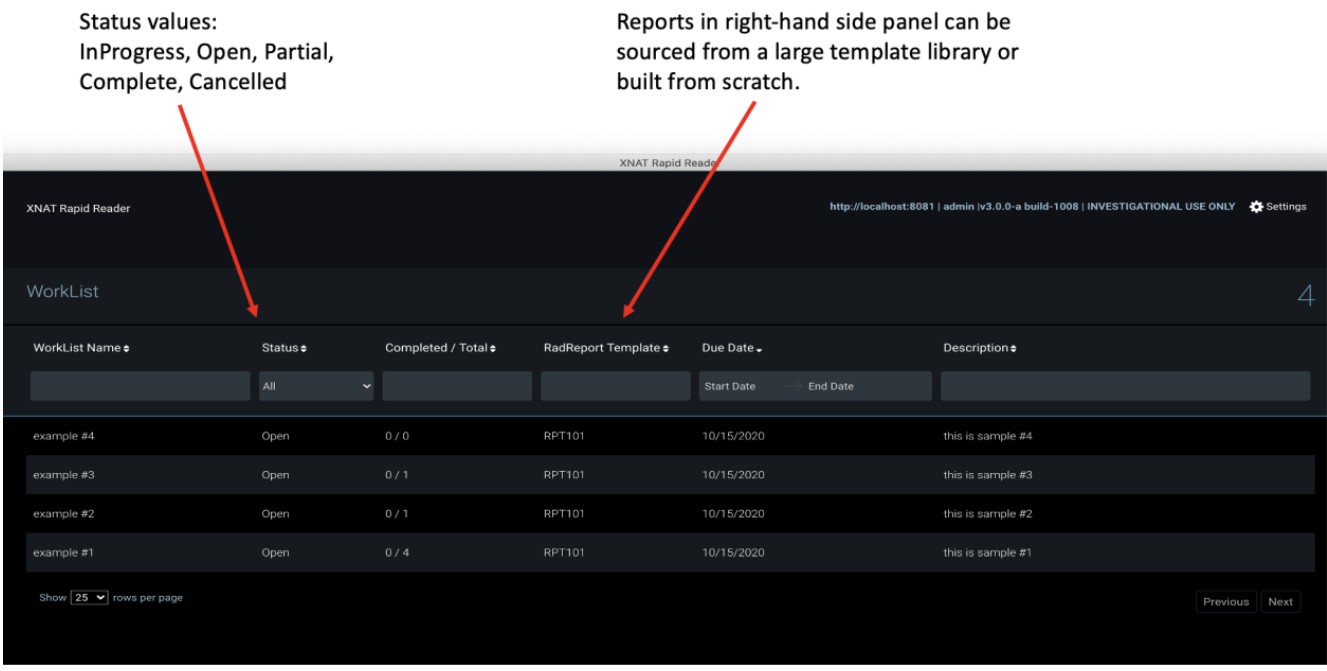

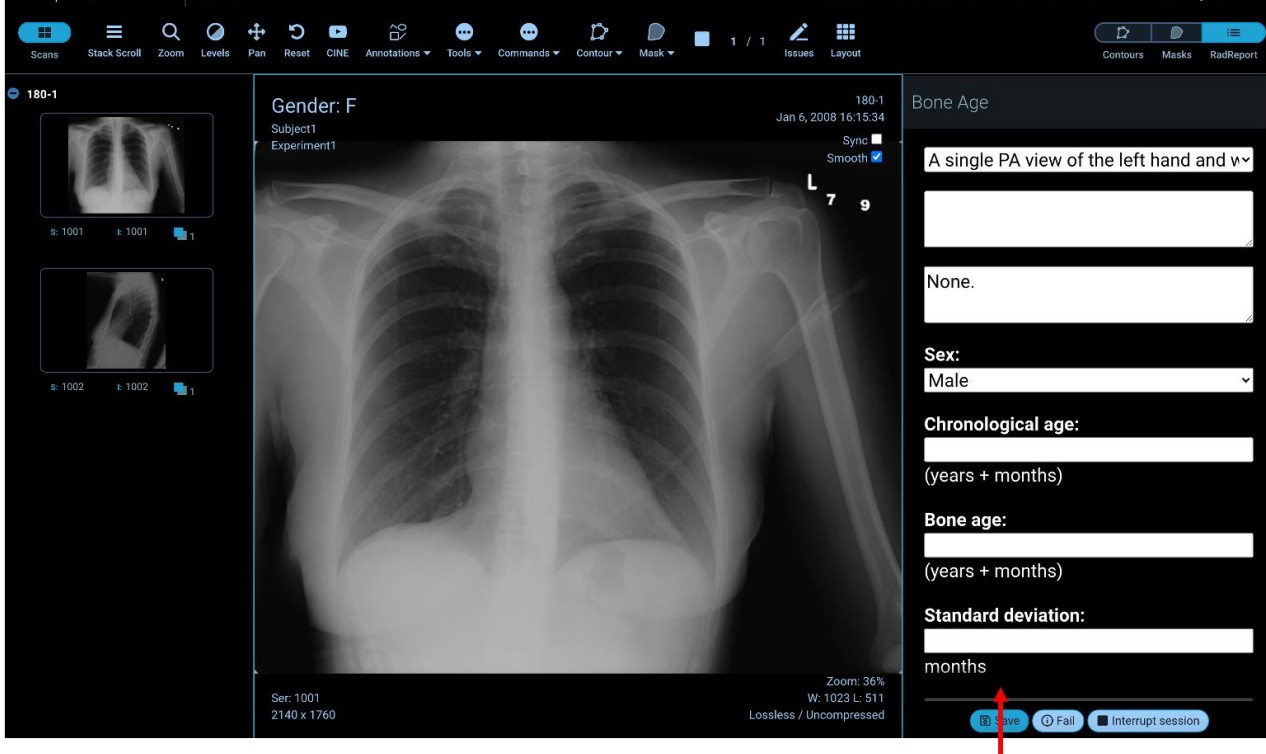

**Figure 7.** Rapid Reader workflow view and modified viewer window illustrating new electronic case report form (eCRF) panel used to render a RadReport template. Note the additional navigation and report controls on the right-hand side of the toolbar. Rapid Reader is currently in development: source code is available on request, but not yet supported by the XNAT team.

## 4. Discussion

Our initial incorporation of the OHIF viewer (Versions 1 and 2) into XNAT satisfied a growing need within the community to visualise data, and new capabilities in the ICR-XNAT-OHIF Version 3 expand further the range of tasks that can be undertaken. They support a more general evolution of the concept of an image data repository from a simple archive towards a tool for active curation and enrichment, via radiologist annotation, of image data in clinical studies. Unlike a traditional PACS, research users can extend both the XNAT and OHIF platforms themselves to add new tools, but at the same time, the viewer shares enough similarity of "look-and-feel" with PACS to be readily accepted by radiologists.

The ability to segment regions of an image flexibly is an essential prerequisite to many quantitative imaging analysis techniques. For example, in dynamic-contrast MRI studies or whole-body MRI diffusion imaging, lesions are identified and regional statistics are computed for tumours.

The work performed so far indicates great potential for the use of the ICR-XNAT-OHIF viewer and this is borne out by expansive download statistics (see Results). Already having thousands of downloads for this new and evolving technology suggests that the integration of the OHIF web viewer with XNAT fills an unmet need in the imaging research community.

OHIF occupies a different and complementary niche in the research ecosystem from "heavyweight", workstation-based research tools such as 3D Slicer [35]. Our framework is a "zero-footprint" web viewer [36]. Such an approach has both advantages and disadvantages:

- There is no overhead of local software installation, the only prerequisite being a standard web browser. Although the user experience is better when running that browser on a high-spec desktop machine, the software performs well even on older computers with more modest capabilities. The viewer can be used very successfully on a tablet and, indeed, using a large tablet together with an appropriate stylus represents a potentially optimum combination for manual annotation of ROIs. Images can even be viewed on a mobile phone, subject to obvious limitations in screen real estate. The web-based approach is well aligned with the needs of "opening" multi-centre trials with standardised software and procedures in organisations with diverse hardware and varying levels of local technical support.

- Computationally intensive tasks (e.g., real-time ML model inference) can be handled server-side, reducing the need for high-performance web clients and bringing advanced techniques within the reach of all centres participating in a trial, thus "democratising" the use of AI.

- Researchers gain immediate access to all images granted by their XNAT permissions, without any need to "import" data into an application or hold them locally on a workstation. For large archives, the accessible data might comprise tens of terabytes, representing hundreds of thousands of subjects.

- By contrast, each time an image is reviewed, it needs to be retrieved from the server in real-time. Depending on data volume and internet speeds, this might be slow, leading to "data-buffering" delays of tens of seconds prior to large 3D images being fully available for viewing. This is a major disadvantage of the zero-footprint design compared with a more traditional desktop application where data are fully resident on a local disk. It currently degrades the viewing experience for radiologists, leading to reluctant uptake in some quarters. The work targets a different use case to a PACS and, at present, OHIF would not be a suitable replacement clinically. However, we envisage future improvements via plausible mitigations to: (i) improve backend efficiency in querying the data (e.g., store each DICOM series as a single compressed file); (ii) employ smarter caching so that images are stored in the most easily displayable representations; (iii) use a "progressive" DICOM codec such as HTJ2K [37,38]; (iv) use server-side image rendering and transmit only the final, rendered image to the browser;

(v) employ novel solutions such as blockchain-enabled distributed storage [39] to bring the data closer to users.

- Browser memory restrictions limit the complexity of data displayed (for example, the number of active 3D DICOM segmentation objects loaded in MPR view). This problem currently has no easy solution.

In the two years since January 2020, we have created five new releases of the XNAT plugin, each increasing the viewer's functionality with substantive new tools. Throughout the process, the NCITA development team has maintained an active dialogue with the XNAT user community, which has resulted in rapid responses to feature requests (e.g., improved ultrasound support), and it also coordinates development closely with the core OHIF and XNAT teams. This agile development would be impossible within a commercial PACS ecosystem.

The roadmap for the future development of the ICR-XNAT-OHIF viewer integration includes:

- enhanced support for radiotherapy objects (e.g., DICOM RT-DOSE);
- enhanced facilities for duplicating ROIs both between DICOM series within the same study and, via registration, between different imaging sessions;
- support for versioning of ROIs;
- enhanced annotation workflows;
- creation of hanging protocols;
- further development of the Rapid Reader;
- full support within XNAT for the OHIF microscopy extension.
- transition to core OHIF-v3.

Finally, a much-requested feature is support for visualising image data in the Neuroimaging Informatics Technology Initiative (NIfTI) format [40]. Whilst decoding and rendering the data themselves is straightforward from a technical perspective, the lack of a mandatory patient-based coordinate system in the NIfTI specification and the lack of mandatory metadata to link NIfTI images representing segmentations with their corresponding NIfTI base images (or the DICOM data from which they were ultimately derived) makes them less-than-ideal candidates for a structured data repository such as XNAT. There has consequently been much debate within the development team as to the best way to ensure data consistency during NIfTI image upload and how to perform adequate quality control on incoming data to avoid situations where the viewer displays overlays that are incorrectly oriented with respect to the base image data ("segmentation flips"). Whilst the Brain Image Data Structure (BIDS) format [41] may provide a way forward, there is not yet a consensus that this is a complete solution.

## 5. Conclusions

The popular OHIF zero-footprint web viewer has been integrated with the XNAT image repository platform. Significant mismatches in the underlying data model between XNAT and DICOM have been overcome, leading to a single-click, seamless visualisation workflow within XNAT. The work described above provides new capabilities, making possible AI-based annotation workflows and data capture via electronic case report forms. While not immediately suitable as a replacement viewer for a busy hospital radiology department, the combination of XNAT and OHIF will streamline the conduct of quantitative imaging clinical trials and related academic studies. Solutions have been proposed for outstanding issues and the joint platform will continue to evolve and increase in functionality.

**Author Contributions:** S.J.D. undertook early development work, managed the project to integrate the OHIF Viewer into XNAT and conceived and wrote the manuscript; M.A.S. was the principal front-end JavaScript developer for Version 3 of the ICR-XNAT-OHIF integration; J.D. was the principal back-end developer of the Java plugin code interfacing with XNAT; J.A.P. was the principal front-end developer for Versions 1 and 2; K.A. wrote the code for the four-up surface-rendering view and contributed numerous helpful technical comments; W.C. wrote the Rapid Reader web application and wrote part of the manuscript; L.E.S. coordinated validation of the AIAA server integration and provided significant technical input to the project; S.A., A.E.H. and B.G. facilitated the integration of NVIDIA's AIAA and wrote part of the manuscript; E.Z. and G.J.H. contributed valuable discussions to the project and represent the OHIF consortium, without which none of the developments reported here would have been possible; E.O.A., E.S. and D.-M.K. provided senior input to the process and are PIs of the NCITA consortium; D.M. manages the core XNAT project and secured funding enabling part of this work. All authors reviewed and commented on the manuscript in progress. The views expressed are those of the authors and not necessarily those of the NIHR, the Department of Health and Social Care or the other funding bodies listed. All authors have read and agreed to the published version of the manuscript.

**Funding:** This study represents independent research supported by the National Institute for Health Research (NIHR) Biomedical Research Centre, the Clinical Research Facility in Imaging and the Cancer Research Network at The Royal Marsden NHS Foundation Trust (RMH) and the Institute of Cancer Research, London (ICR), as well as the National Institute of Health Research (NIHR) Cambridge Biomedical Research Centre (BRC-1215-20014). Staff effort for this project was supported by CRUK funding C4278/A27066 for the National Cancer Imaging Translational Accelerator (NCITA) and for the Cancer Research UK (CRUK) Cambridge Centre [C9685/A25177], as well as National Cancer Institute (NCI) grants 1U24CA204854 and 1U24CA199460. This project was made possible in part by grant 2020-225168 from the Chan Zuckerberg Initiative DAF, an advised fund of Silicon Valley Community Foundation, and has also benefited from historical support of the Cancer Imaging Centre at the RMH and the ICR from Cancer Research UK (CRUK) and the Engineering and Physical Sciences Research Council, in association with Medical Research Council and Department of Health C1060/A10334, C1060/A16464. E Ziegler is supported by Radical Imaging LLC and via NCI grant R01CA235589 as a subcontract to Novometrics LLC.

**Institutional Review Board Statement:** Not applicable.

**Informed Consent Statement:** Not applicable.

**Data Availability Statement:** Not applicable.

**Acknowledgments:** Although, in the years since the inception of the project, numerous commercial and academic ventures have incorporated the OHIF Viewer, few such examples were available at the outset. The authors are thus grateful for initial hints and preliminary expertise gained by Amin EL-Rowmeim, Department of Radiology at Memorial Sloan Kettering Cancer Centre. The data for Figure 4 were made available by kind permission of Ben Glocker (Imperial College London) Alexandra Taylor and David Bernstein (Royal Marsden Hospital).

**Conflicts of Interest:** The funders had no role in the design of the study; in the collection, analyses, or interpretation of data; in the writing of the manuscript, or in the decision to publish the results. E.O.A. is a member of the Editorial Board of Tomography. G.J.H. is a member of Novometrics LLC and IQ Medical Imaging LLC and an advisor for Fovia Inc. E.S. is co-founder and shareholder of Lucida Medical Ltd. L.E.S. has received consulting fees from Lucida Medical Ltd.

## Abbreviations

| | |
|---|---|
| AIAA | Artificial Intelligence Assisted Annotation |
| AIM | Annotation and Imaging Markup |
| API | Application Programming Interface |
| DICOM | Digital Imaging and Communications in Medicine |
| IOD | Information Object Definition |
| MONAI | Medical Open Network for AI |
| NIfTI | Neuroimaging Informatics Technology Initiative |
| OHIF | Open Health Imaging Foundation |
| PACS | Picture Archiving and Communications Systems |

| P10 | DICOM Part 10 |
| REST | Representational State Transfer |
| ROI | Region-of-Interest |
| STL | Standard Triangle Language (or Standard Tesselation Language) |
| UI | User interface |
| VR | Value Representation |
| XNAT | eXtensible Neuroimaging Archive Toolkit (Note that XNAT was originally an acronym with this definition. However, recognition by the core XNAT team that its remit runs far beyond neuroimaging has led to the current practice of regarding XNAT as a name, not an acronym any longer requiring definition) |

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
