# Peer review of "Integrating the OHIF Viewer into XNAT: Achievements, Challenges and Prospects for Quantitative Imaging Studies"

_tomography, doi:10.3390/tomography8010040_

Round 1

Reviewer 1 Report

In this review article, the authors have discussed about describe the background to and implementation of an integration of the Open Health Imaging Foundation (OHIF) Viewer into the XNAT environment. Additional suggestions for improvements are as follows:

  1. The authors should also clearly highlight the novelty as few articles have already been previously published on similar topic.
  2. The conclusion section is appropriate and provides enough and useful information. However, it should be elaborated by adding information about clinical limitations and future directions.
  3. The authors should provide their own justification and relevance of the study. This will help the readers to understand the importance of the paper.
  4. All sections of manuscript should be checked in terms of spell and grammar mistakes.

Overall, this review is a sort of list of known facts, but I'd like that authors express their opinions or critical point of view on the literature. Several points would need to be addressed. Most importantly, I find several passages of your current manuscript very descriptive and thus, it would be important to focus on the conceptual interpretation of findings.  It is important that the review does not end up being an annotated bibliography (i.e. a list of recent findings with no real context or analysis). A review should not only be useful for finding papers, but also push our conceptual understanding forward. As you read through your manuscript, please ensure that the implications and take-home messages are clear. Each paragraph should end with a summary and synthesis that links to the main message of your article. This will also help ensure a better flow of the manuscript, connecting the arguments.

Author Response

Reviewer #1 general comments

Taken together this reviewer’s comments argue for more clarity in the role of each section. We appreciate the opportunity to better “signpost” our work and hope that the revised version achieves this. In response to the points below, we have rewritten the Introduction, adding subsection headings, and augmented both the Discussion and Conclusions.

  • Reviewer comment: Overall, this review is a sort of list of known facts… It is important that the review does not end up being an annotated bibliography (i.e. a list of recent findings with no real context or analysis). A review should not only be useful for finding papers, but also push our conceptual understanding forward.
  • Response: Whilst necessarily involving some element of review, this work is not primarily intended to be a “review article” but, rather, a report of novel research. We now make this clear in a summary of the paper, which we have relocated to the second paragraph of the Introduction.

We do provide extensive background descriptions of some technical aspects (e.g., the XNAT and DICOM information models), as we do not expect readers of the QIN special issue necessarily to be familiar with this material. These are “known facts”, but we provide a novel commentary with fresh insights compared with the description in ref. [1]. Conversely, for those outside of QIN, and particularly those unaware of the UK context, it is important to understand how the role of the quantitative imaging community has been important in facilitating these developments.

We note, too, that some aspects of the work presented here are “known facts” simply because they are embodied in our software, which is now regularly used by hundreds of researchers worldwide. However, this manuscript represents the first formal record in the academic literature of the new developments described here.    

  • Reviewer comment: I'd like that authors express their opinions or critical point of view on the literature.

Response: As pointed out above, it is not our aim in this article to produce a comprehensive, critical review of the literature, but to place our developments in context. Nevertheless, following the reviewer suggestions, we expand the Discussion to comment more critically on the relative merits of thin clients like OHIF and workstation-based applications, and we make the advantages and disadvantages of our system clearer.

  • Reviewer comment: Several points would need to be addressed. Most importantly, I find several passages of your current manuscript very descriptive and thus, it would be important to focus on the conceptual interpretation of findings.

Response: We have improved the Introduction and added headings to highlight the concepts involved. We also reformatted the Discussion to highlight via bullet points the different concepts involved. However, we are unclear as to what the reviewer means by “conceptual interpretation of findings”.   

  • Reviewer comment: As you read through your manuscript, please ensure that the implications and take-home messages are clear. Each paragraph should end with a summary and synthesis that links to the main message of your article. This will also help ensure a better flow of the manuscript, connecting the arguments.

Response: We hope that the revised version of the manuscript has made the take-home messages clearer. We respectfully disagree with this reviewer’s comment that each paragraph should end with a summary and synthesis; this would be excessive and increase the length of the manuscript in an unwarranted fashion. However, we have modified the Conclusion to make the take-home messages much clearer and more direct.

Reviewer #1 specific comments

  • Reviewer comment: The authors should also clearly highlight the novelty as few articles have already been previously published on similar topic.

Response: Following the suggestion of Reviewer #4, we add more background on other web viewers in the Introduction. We now explicitly draw attention to the novel and unique features of this work on p. 6.  

  • Reviewer comment: The conclusion section is appropriate and provides enough and useful information. However, it should be elaborated by adding information about clinical limitations and future directions.

Response: We have added relevant content to the Conclusions.

  • Reviewer comment: The authors should provide their own justification and relevance of the study. This will help the readers to understand the importance of the paper.

Response: We believe that the new summary text on p. 4 and the additional text on p. 6 now fulfil this request.

  • Reviewer comment: All sections of manuscript should be checked in terms of spell [sic] and grammar mistakes.

Response: The original manuscript was carefully checked, and we apologise for any typos or errors that crept into the text. The first author is a native English speaker, so we are surprised at the comments of Reviewers #1 and #2 that “moderate” changes to the English are required.

Reviewer 2 Report

This paper provides a review for integrating the OHIF Viewer into XNAT. Furthermore, the authors promoted some ideas to handle the current challenges. I believe this paper is valuable for upcoming researches focused on OHIF, XNAT and their integration. However, I have a few comments and suggestions which hopefully would help perfecting the manuscript.

  • A description of setions should be added in Introduction.
  • I suggest that authors re-structure their papers. For example, the development history and Initial challenges should be included in the section of Backgroud….
  • The authors may consider a flow chart to describe the integration of OHIF Viewer and HNAT, This would make it easier for the reader to understand,

Author Response

  • Reviewer comment: A description of setions [sic] should be added in Introduction.

Response: This has now been done. See the responses to Reviewer #1 above.

  • Reviewer comment: I suggest that authors re-structure their papers. For example, the development history and Initial challenges should be included in the section of Backgroud….

Response: As described in the response to Reviewer #1, we have restructured the paper. We disagree that the development history and initial challenges should be included in the background. These all form part of the novel work described in this manuscript. The current incarnation of the OHIF viewer integration is the result of a five-year programme of development that is being reported in the academic literature for the first time here.   

  • Reviewer comment: The authors may consider a flow chart to describe the integration of OHIF Viewer and HNAT, This would make it easier for the reader to understand,

Response: We believe that a flow chart representing the integration would not contain information significantly different from the timeline of Figure 1.

Reviewer 3 Report

1) This paper appears to be an interesting contribution to research on ‘informatics software platform’ ‘for Quantitative Imaging Studies’;

2) Its academic and scientific values make it suitable for the international, open access and peer-reviewed journal Tomography (ISSN 2379-139X);

3) However, there are a few minor writing and formatting issues that should be addressed by the authors. See, for instance, the “examples” listed below (underlined text):

 “1. Introduction”:

  • CHECK TEXT, a ‘period’ seems to be missing: ‘This is typically achieved by equipping platforms with Representational State Transfer (REST) [2]???’;

“2.2. Initial challenges”:

  • CHECK TEXT, some ‘commas’ seem to be missing: ‘XNAT’s concept of a “project” ??? the top level of the storage hierarchy in the XNAT repository structure ??? has no direct equivalent in DICOM.’;

“5. Conclusions”:

  • SECTIONS, CHECK NUMBERING.

Author Response

  • Reviewer comment: CHECK TEXT, a ‘period’ seems to be missing: ‘This is typically achieved by equipping platforms with Representational State Transfer (REST) [2]???’;

Response: It appears that the definition of acronyms here and the insertion of a reference to Fielding’s REST paper have got in the way of the comprehensibility of the text. Without these the sentence becomes clear and needs no full stop.

This is typically achieved by equipping platforms with REST APIs, thus enabling integration with a diverse range of end user tools.

This issue has been ameliorated by moving the citation to the end of the sentence in the revision.

  • Reviewer comment: CHECK TEXT, some ‘commas’ seem to be missing: ‘XNAT’s concept of a “project” ??? the top level of the storage hierarchy in the XNAT repository structure ??? has no direct equivalent in DICOM.’;

Response: The original text used em-dashes at the points where ??? appears in this comment and no commas were intended.

  • Reviewer comment: SECTIONS, CHECK NUMBERING.

Response: Both the original and this revision were deliberately left un-numbered, as no references from one section to another were needed. We leave it to the journal copy-editors to determine whether section numbers are needed to conform to the house style.

Reviewer 4 Report

The reviewer had minor comments related to the presentation of the article as follows.

  1. The title of the article should be reformatted to make it easier to read, the word "Challenges" should be presented in one line.
  2. The “ Introduction” should move to the next page.

General comments to the structure, meaning, results, etc. of the article: The article summarizes the formation and development of the XNAT toolkit from 2015 to 2021 and includes an introduction to the features of this toolkit.

  1. The abstract of the article has not really highlighted the key influence of XNAT in researches and medical applications. The reviewer thinks that the authors should add more the results as numbers because the numbers can more easily convince the reader (i.e. ….’’significant interest (3317 downloads of the various plugins).
  2. In the introduction " This article describes the integration by ICR and NCITA of the OHIF Viewer into XNAT....", the authors should highlight the importance and main contributions in this article.
  3. At the end of the introduction, the authors should add a small paragraph to explain the layout of the article so that it is easier for readers to follow.
  4. The “Materials and methods” and “Results” sections are mainly about the integration capabilities and functions of the user interface. The article has not mentioned the requirements of hardware (i.e. personal computer). The customers always consider that hardware requirements are one of the criteria for selecting and using the software. The author team should have more comments, assessments and recommendations about the hardware compatibility of this software. because it is directly related to the efficiency of the software.

Author Response

Reviewer #4 formatting comments

  • Reviewer comment: The title of the article should be reformatted to make it easier to read, the word “Challenges” should be presented in one line.
  • Reviewer comment: The “Introduction” should move to the next page.

Response: We did not understand these. The title of the original manuscript did not split the word “challenges” and the Introduction started on a new page in the version originally sent by the authors. Could it be that the version that the reviewer received contained additional formatting performed by the journal staff?

Reviewer #4 specific comments

  • Reviewer comment: The abstract of the article has not really highlighted the key influence of XNAT in researches and medical applications. The reviewer thinks that the authors should add more the results as numbers because the numbers can more easily convince the reader (i.e. ….’’significant interest (3317 downloads of the various plugins).

Response: During the month that the manuscript was in review, a further 400 downloads occurred, emphasising the popularity of the tool. We add the download statistic to the abstract. However, it is beyond the scope of the current work to discuss the “key influence” of XNAT as a whole. Instead, we insert a reference to XNAT’s role in the Human Connectome Project as one example.

  • Reviewer comment: In the introduction " This article describes the integration by ICR and NCITA of the OHIF Viewer into XNAT....", the authors should highlight the importance and main contributions in this article.

Response: The text highlighted by the reviewer has now moved location to the second paragraph of the Introduction on p. 4, and we believe the novel contributions of this piece of work have been better highlighted throughout the text. Please see the responses to Reviewer #1.

  • Reviewer comment: At the end of the introduction, the authors should add a small paragraph to explain the layout of the article so that it is easier for readers to follow.

Response: We believe that the new paragraph 2 of the Introduction serves this purpose. See 4.2 above.

  • Reviewer comment: The “Materials and methods” and “Results” sections are mainly about the integration capabilities and functions of the user interface. The article has not mentioned the requirements of hardware (i.e. personal computer). The customers always consider that hardware requirements are one of the criteria for selecting and using the software. The author team should have more comments, assessments and recommendations about the hardware compatibility of this software. because it is directly related to the efficiency of the software.

Response: This is a very helpful comment and we had not adequately addressed this issue previously. The revision contains an extensive bullet point on p. 15 detailing hardware compatibility.

Reviewer 5 Report

The authors present the integration of the OHIF Viewer into XNAT. The work enables, for example, a browser-based segmentation of medical data, and can be used as alternative to installer-based research tools such as 3D Slicer. The authors should mention (and maybe briefly discuss) other browser-based segmentation tools for (bio-)medical data, like GradioHub [1], StudierFenster [2] and Biomedisa [3].

[1] Abid A, Abdalla A, Abid A, Khan D, Alfozan A, Zou J. An online platform for interactive feedback in biomedical machine learning. Nature Machine Intelligence. 2020 Feb;2(2):86-8.
[2] Wild D, et al. A Client/Server Based Online Environment for Manual Segmentation of Medical Images. In CESCG 2019: 23rd Central European Seminar on Computer Graphics 2019, pp. 1-8.
[3] Lösel PD, van de Kamp T, Jayme A, Ershov A, Faragó T, Pichler O, Jerome NT, Aadepu N, Bremer S, Chilingaryan SA, Heethoff M. Introducing Biomedisa as an open-source online platform for biomedical image segmentation. Nature communications. 2020 Nov 4;11(1):1-4.

Author Response

  • Reviewer comment: The authors present the integration of the OHIF Viewer into XNAT. The work enables, for example, a browser-based segmentation of medical data, and can be used as alternative to installer-based research tools such as 3D Slicer. The authors should mention (and maybe briefly discuss) other browser-based segmentation tools for (bio-)medical data, like GradioHub [1], StudierFenster [2] and Biomedisa [3].

Response: We thank the reviewer for these suggestions and have incorporated them into a new section on p. 6 listing other web viewers.

Round 2

Reviewer 4 Report

The manuscript was revised following the revirewer comments. No more questions.